Follow Up

# SOX2 is a dispensable modulator of NUT carcinoma oncogenesis in mice

Chenxiang Luo[1,2,6], Dejin Zheng[1,2] (iD), Ahmed Elnegiry[1,2,7], Sheetal Bhatara[3] (iD), George I Mias[2,4] (iD), Mayra F Tsoi[5] (iD), Jiyang Yu[3], Bin Gu[1,2] (iD)

NUT carcinoma (NC) is an aggressive malignancy driven by *BRD4::NUTM1* and other *NUTM1* fusion oncogenes. BRD4::NUTM1 aberrantly activates transcription factors (TFs) associated with basal progenitor cells of stratified epithelium, resulting in a poorly differentiated squamous cell carcinoma (SCC) phenotypes. Among these TFs, SOX2 has been proposed as a critical driver. However, its role in NC initiation and progression has not been investigated in vivo. Using a genetically engineered mouse model that faithfully recapitulates human NC, we performed lineage-specific conditional deletion of *Sox2* in both squamous and non-squamous tissues during NC oncogenesis. We found that SOX2 is dispensable for NC initiation and progression, and that tumors lacking SOX2 retain characteristic histological features and expression of key oncogenic drivers, including BRD4:: NUTM1, MYC, and TP63. Bulk RNA sequencing revealed only modest transcriptional changes in SOX2-deficient tumors, primarily affecting metabolic and biosynthetic pathways, without disrupting core oncogenic programs. These findings challenge the assumption that SOX2 is universally required for NC oncogenesis and highlight the autonomy of BRD4::NUTM1 in establishing and maintaining the NC phenotype. Our results suggest that SOX2 is dispensable for NC and redirect therapeutic focus toward BRD4::NUTM1 and its chromatin remodeling dependencies.

## Introduction

NUT carcinoma (NC) is an aggressive solid tumor driven by the *BRD4::NUTM1* and other *NUTM1* fusion genes (1, 2). Currently, no effective treatment is available for NC. A diagnosis of NC is associated with a near-universal fatal outcome (median overall survival (OS) is 6.5 mo) (3, 4). Though perceived as rare, NC is likely substantially underdiagnosed because of its ability to arise in diverse anatomical sites across the body and its presentation with a poorly differentiated, ambiguous histological phenotype (3, 5). More cases are being diagnosed and identified from retrospective cohorts with the advent of specific diagnostic techniques, including NUTM1 immunohistochemistry using specific antibodies, and unbiased genome and transcriptome sequencing. A recent estimation suggests about 1,400 new cases per year in the United States (6, 7), a figure that is expected to increase with growing diagnostic awareness and research efforts worldwide.

Mechanistic studies that dissect the gene regulatory network have provided critical insights into tumor initiation and progression, leading to the development of effective treatments for both common and rare tumors, and similar breakthroughs are anticipated for NC. For decades, investigation on NC biology relied on human NC patient–derived cell lines and their xenograft models, which have elucidated pivotal oncogenic pathways like the BRD4–NUT–p300 axis and enabled early therapeutic strategies, such as bromodomain and extra-terminal motif (BET) protein inhibitors (8, 9, 10, 11, 12). However, almost all cell lines are derived from advanced and often heavily treated tumors and removed from their native tissue environment, limiting their capacity to model the early tumorigenic events and natural evolution of NC. Recently, two concurrently published NC genetically engineered mouse models (GEMMs) have demonstrated that BRD4::NUTM1 is sufficient as a sole oncogenic driver to initiate aggressive tumors resembling human NC (hNC) (13, 14). One model employs a *Brd4:: hNUTM1* knock-in allele activated by the FLEx system and Sox2-CreERT2, resulting in widespread fusion expression and consistent tumor formation in the esophagus, including the gastroesophageal junction (13). The other is the NUT carcinoma translocator (NCT) model (MMRRC-071753-MU), developed by our group. Leveraging Cre-lox–mediated interchromosomal recombination, the NCT model induces a chromosome translocation t(2; 17) syntenic to the human t(15; 19) and generates a *Brd4::Nutm1* fusion gene (Fig 1A,

[1]Department of Obstetrics, Gynecology and Reproductive Biology, College of Human Medicine, Michigan State University, East Lansing, MI, USA [2]Institute for Quantitative Health Science and Engineering, Michigan State University, East Lansing, MI, USA [3]Computational Biology, St. Jude Children's Research Hospital, Memphis, TN, USA [4]Department of Biochemistry and Molecular Biology, College of Natural Science, Michigan State University, East Lansing, MI, USA [5]Department of Pathobiology and Diagnostic Investigation, College of Veterinary Medicine, Michigan State University, East Lansing, MI, USA [6]Home Institution: Center for Reproductive Medicine and Department of Gynecology and Obstetrics, The First Affiliated Hospital, Sun Yat-Sen University, Guangzhou, China [7]Home Institution: Department of Cytology and Histology, Faculty of Veterinary Medicine, Aswan University, Aswan, Egypt

Correspondence: gubin1@msu.edu

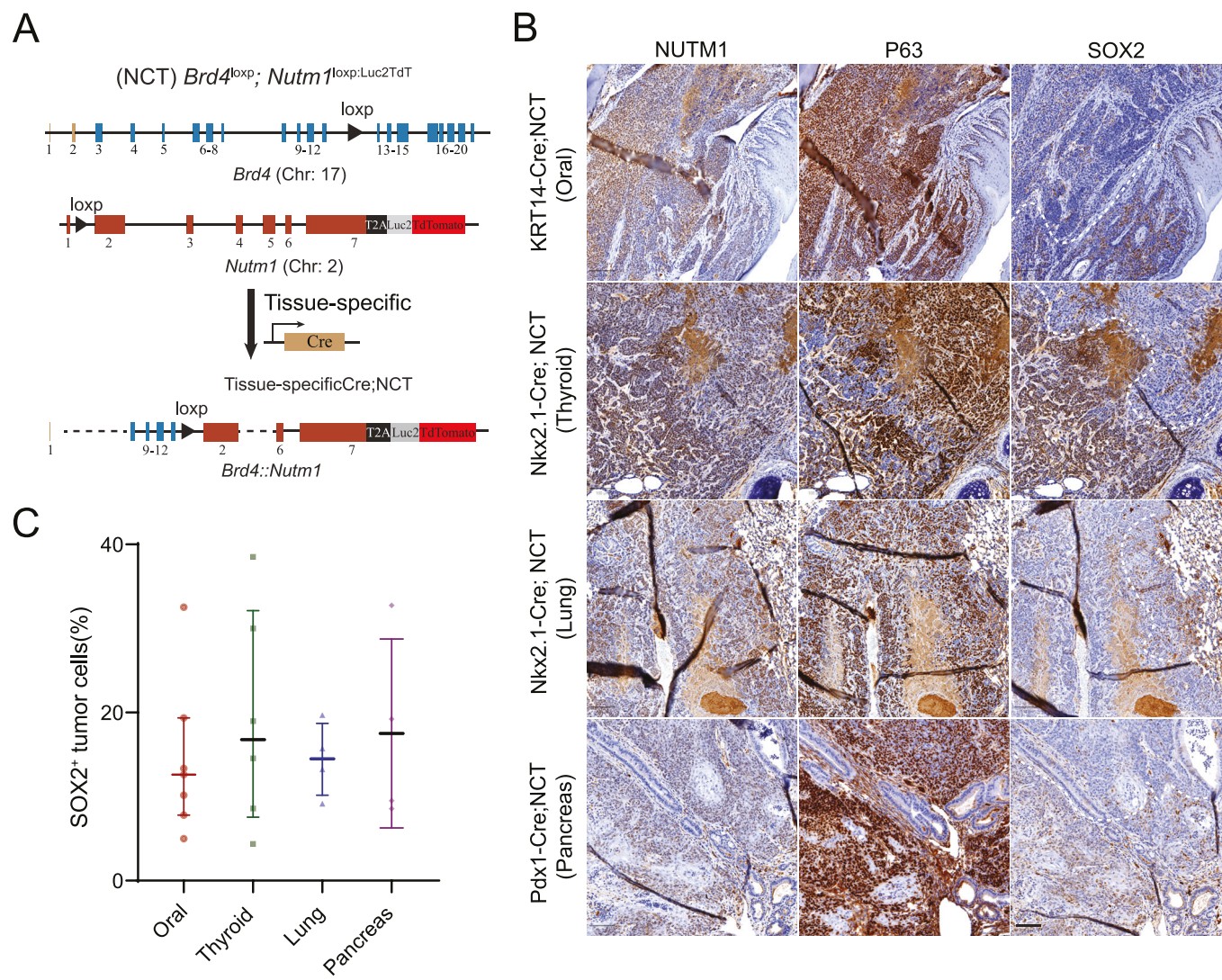

**Figure 1. SOX2 is heterogeneously expressed in mouse and human NUT carcinoma (NC).**
**(A)** Schematic of lineage-specific NCT-based genetically engineered mouse models (GEMMs) used to generate NC tumors from the oral cavity (KRT14-Cre), thyroid and lung (Nkx2.1-Cre), and pancreas (Pdx1-Cre) (reproduced from reference 14 under CC BY 4.0). **(B)** Immunohistochemistry (IHC) for SOX2 in mouse NC tumors. White dashed lines demarcate the interface between SOX2⁻ and SOX2⁺ tumor regions, with SOX2⁺ cells often localized near adjacent morphologically normal tissue. **(B)** Scale bar: 100 μm. *Note: The SOX2 panel in* (B) *(first row) and the SOX2 image in* Fig 3A *(bottom right subpanel) were derived from the same whole-slide scan of the same tissue section, shown at different magnifications and regions of interest.* **(C)** Quantification of SOX2⁺ cells in tumors from different tissue origins. n = 7 (oral), 6 (thyroid), 4 (lung), 4 (pancreas). Bars represent the median with an interquartile range.

reproduced from reference 14 under CC BY 4.0). Because of the inherent low efficiency of interchromosomal recombination, the resulting fusion mutation event is rare and at clonal frequency, uniquely recapitulating the sporadic, clonal origin of NC initiation in hNC, in contrast to the knock-in model in which all target cells uniformly express the fusion. Lineage-specific induction of this event yields highly aggressive tumors at ~100% penetrance, faithfully phenocopying the diverse anatomical distribution, the poorly differentiated squamous cell carcinoma (SCC) phenotype, and molecular markers (p63, c-MYC, and SOX2) observed in human NC. Thus, these NC GEMMs offer an ideal and powerful platform that now enables new opportunities to investigate the mechanisms of NC initiation and progression.

NUT carcinoma is primarily considered an epigenetically driven cancer in which the BRD4–NUT–p300 axis promotes widespread chromatin hyperacetylation and activates the expression of SOX2, MYC, TP63, and likely other transcription factors to drive NC tumorigenesis and SCC phenotype (7, 8, 9). Among these, SOX2 is a powerful TF critical for the self-renewal, regeneration, and differentiation control of stratified epithelial basal progenitor cells and other stem cell types, including embryonic stem cells and neural stem cells (15, 16). Furthermore, SOX2 has been shown to be required for the sustenance of the malignancy properties of human SCC cell lines in vitro, and the initiation and progression of SCCs from the skin and esophagus in vivo (15, 17, 18, 19). Indeed, SOX2 has been proposed as a potential stem cell marker of NC. In

human NC cell lines, RNAi-based SOX2 knockdown induces cellular differentiation and impairs tumor sphere and colony formation, albeit with only moderate effects on proliferation (20). Moreover, the role of SOX2 in NC has also been implicated in the NC GEMM generated by Durall et al, in which mNC tumors arise from *Sox2*-expressing cells (13). However, hNCs can arise from not only tissues with a large number of SOX2-positive cells such as paranasal sinuses and oral mucosa (21, 22), but also tissues that generally lack any SOX2-positive cells, such as the distal lung tissues, thyroid glands, and pancreas (3, 23, 24). This pattern is recapitulated by the NCT-based mouse models (14). These findings raise fundamental questions about whether SOX2 acts as a universal driver or context-dependent marker in NC, and whether it is essential for tumor initiation and maintenance. To address this, we performed conditional *Sox2* knockout in multiple NCT model lineages and found that SOX2 is dispensable for the initiation and progression of NC, regardless of whether the tissue of origin normally expresses *Sox2*. Transcriptomic analysis revealed only modest changes, primarily affecting metabolic pathways, with no disruption of core oncogenic programs in SOX2-deficient tumors. Our findings challenge the assumed universal dependency on SOX2 in NC and suggest that cautions should be taken for pursuing SOX2-targeting agents as a monotherapy for NC.

# Results

## SOX2 is expressed in subpopulations of NC cells with high inter-tumor variability

SOX2 has been proposed as a downstream target activated by BRD4::NUTM1 in NC (8, 20). To assess its expression, we performed SOX2 immunohistochemistry on mouse NC (mNC) tumors derived from multiple tissue lineages using NCT models driven by different Cre lines (14). Serial neighboring sections were analyzed from mNCs originating in the oral cavity (KRT14-Cre; NCT model), thyroid (Nkx2.1-Cre; NCT model), lung (Nkx2.1-Cre; NCT model), and pancreas (Pdx1-Cre; NCT model), encompassing the major anatomical sites where human NC has been reported. Unlike other NC markers, such as BRD4::NUTM1 and p63, which are diffusely expressed across most tumor cells, SOX2 is restricted to discrete subpopulations, often located near the tumor margin adjacent to the morphologically normal tissue (Fig 1B). Furthermore, the frequency of SOX2-expressing cells within mNCs is highly variable, with no clear correlation with the tissues of origin (Fig 1C). SOX2 is not routinely assessed in clinical NC diagnosis and has been sparsely reported in literature. However, in limited studies where SOX2 was examined, it is highly expressed in most cases (4/5) in one study (20), but expressed only in subpopulations of NC cells in two others (20, 25, 26). To further investigate the relevance of this pattern in human NC, we interrogated the RNA-seq data of 12 human NC cell lines. These cell lines represent diverse patient demographics, including age range and tumor locations of human NC (27). High variability of *SOX2* expression was observed, with a median expression level of 18,489 transcripts and an interquartile range (IQR) of 14,742 (Fig S1). As a comparison, the expression of the *BRD4::NUTM1* fusion gene is

consistent among the cell lines (median: 3,865; IQR: 1,964). These findings demonstrate that SOX2 is not universally expressed in NC cells within tumors and cell lines, implicating that it is not universally required for NC survival. However, it may instead play a role in NC initiation or define a small, functionally important stem-like subpopulation. To test these hypotheses, we conducted conditional *Sox2* knockout experiments to evaluate its role in NC development.

## SOX2 is not required for mNC initiation and progression from Krt14-expressing tissues

The NCT mouse model induces low-frequency, sporadic Brd4::Nutm1 chromosomal translocations that drive NC formation via constitutively active Cre recombinase expressed in defined lineages. A tdTomato-Luc2 reporter knocked in downstream of the testis-specific *Nutm1* gene allows in vivo monitoring of tumor initiation and growth by bioluminescence imaging (14). To assess the functional requirement of SOX2 in mNC development, we introduced a floxed *Sox2* allele into this system, allowing Cre-mediated deletion of *Sox2* in the specific cell population from which mNC arises (Fig 2A).

Because *Sox2* deletion in respiratory epithelium causes perinatal lethality (28), we started with the KRT14-Cre–driven mNC model, which induces recombination in ectoderm-derived epithelial basal cells, mainly in oral mucosa, skin, and salivary glands (29). As shown in Fig 2B and C, similar to *Sox2*-WT controls (KRT14-Cre; NCT$^{+/−}$), the KRT14-Cre; NCT$^{+/−}$;*Sox2*$^{fl/fl}$ mice develop tumors at 100% penetrance. Tumors primarily arose in the oral mucosa, with additional tumors in the skin or salivary glands in about half of the mice. Tumor growth kinetics were comparable between genotypes, with rapid progression leading to early mortality (Fig 2D). Histologically, *Sox2*-deficient mNCs resembled WT tumors, displaying features of poorly differentiated squamous cell carcinoma with abrupt keratinization, consistent with classic NC morphology (Fig 2E). IHC confirmed complete loss of SOX2 expression in knockout tumors, whereas other canonical NC markers including BRD4::NUTM1, p63, and MYC remained robustly expressed (Fig 2F and G). In addition, both genotypes showed similar high levels of Ki-67 expression, suggesting SOX2 loss did not impair tumor growth, consistent with the survival curve.

To rule out the possibility that *Sox2* was not effectively deleted in tumor cells (a potential concern in GEMMs where rare escapees may give rise to tumors), we examined whole-head sections by IHC. In control mice, nuclear SOX2 staining was present in oral tumors, adjacent oral mucosa, and nasal epithelium (Fig 3A). In *Sox2*-deficient mice, SOX2 staining was absent in both the oral tumors and the KRT14-Cre–targeted oral mucosa, confirming effective gene deletion. In contrast, robust SOX2 staining was retained in nasal epithelium where KRT14-Cre is inactive, demonstrating both successful conditional knockout and antibody specificity. Although IHC or IF cannot completely exclude the presence of a few NC cells retaining the *Sox2* allele, the genetic configuration of this GEMM strongly supports efficient deletion. In this model, the same Krt14-Cre event that induces the *Brd4::Nutm1* translocation simultaneously excises the *Sox2* fl alleles, and the frequency of floxed excision far exceeds that of chromosomal translocation. Because

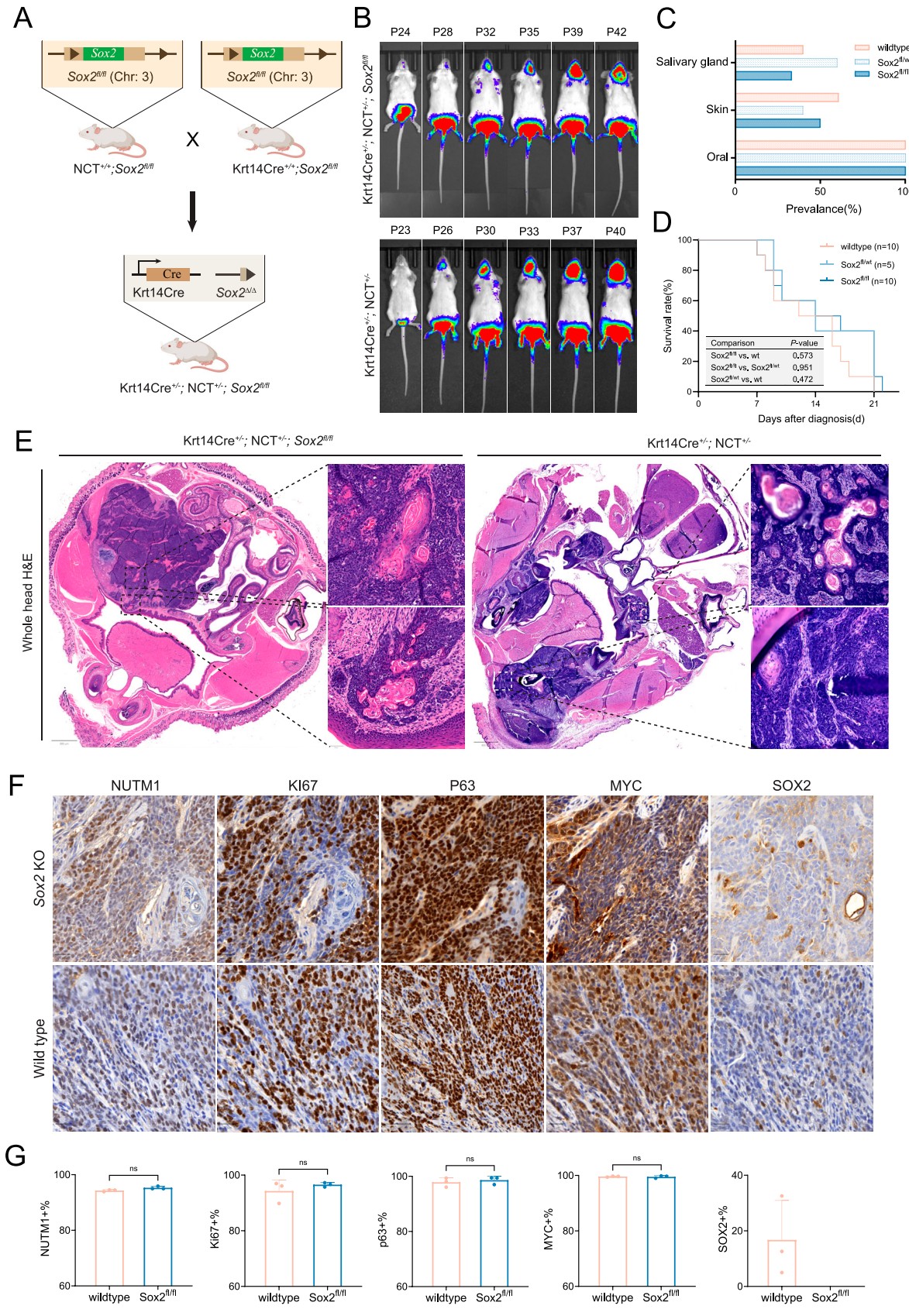

Krt14-Cre remains active in NC cells, any residual floxed alleles would be deleted during tumor progression, making it highly improbable that a *Brd4::Nutm1*-positive tumor retains intact *Sox2*.

Importantly, SOX2+ progenitors are naturally present in oral mucosa and salivary glands, but not in the interfollicular epidermis, which lacks SOX2 expression (18, 30, 31). Despite this, approximately half of both *Sox2* WT and knockout mice developed mNCs in the skin, with tumors exhibiting indistinguishable histology, consistent with poorly differentiated squamous cell carcinoma with abrupt keratinization (Fig 3B). These findings reinforce that SOX2 is dispensable for mNC development not only in SOX2-expressing tissues but possibly also in tissues that do not express SOX2 under normal physiological conditions. However, monitoring skin tumor development in this model was challenging. As the KRT14-Cre; NCT system predominantly initiates tumors in the oral cavity, tumor burden on this site often impairs drinking, feeding, and breathing before skin tumors become externally detectable or measurable. This limitation hinders accurate assessment of tumor latency and penetrance in the skin compartment.

To overcome this, we next designed conditional *Sox2* knockout experiments in the pancreas that do not normally express SOX2, allowing us to test whether ectopic SOX2 activation is necessary for NC oncogenesis, without the confounding effects of early-onset oral tumors.

### Loss of *Sox2* does not impair mNC initiation or progression in *Sox2*-absent progenitors from the pancreas

SOX2 is a known regulator of progenitor cells in ectodermal and anterior foregut-derived tissues such as the oral mucosa, skin, salivary gland, and esophagus (30). However, SOX2 is neither expressed nor required during the development of posterior foregut-derived structures such as the pancreas and bile duct, which arise from Pdx1-expressing progenitors (29). To test whether SOX2 is required for *Brd4::Nutm1*-driven tumorigenesis in this context, we deleted *Sox2* in the Pdx1-Cre; NCT model. Despite arising from SOX2-negative progenitors, the Pdx1-Cre; NCT+/−;*Sox2*fl;fl mice developed mNCs in the pancreas at 100% penetrance, comparable to the *Sox2* WT controls. The tumor growth kinetics (Fig 4A and B), survival curve (Fig 4C), and disease progression were also indistinguishable between genotypes. Notably, most mice reached humane endpoints within 3 wk of tumor detection by BLI, mirroring the rapid progression observed in the KRT14-Cre; NCT models, regardless of the *Sox2* status. Histologically, *Sox2*-deficient tumors in the pancreas exhibited classic features of poorly differentiated squamous cell carcinoma, similar to *Sox2* WT tumors (Fig 4D). IHC analysis of serial sections confirmed complete loss of SOX2 in knockout tumors, without affecting the expression of key

NC markers including BRD4::NUTM1, p63, and MYC. The percentage of Ki-67–positive tumor cells was also comparable between groups, indicating that SOX2 loss does not impair proliferative capacity (Fig 4D and E). No apparent developmental growth or pancreatic/bile duct defects were observed in the Pdx1-Cre; NCT+/−;*Sox2*fl;fl mice. Together, these data demonstrate that SOX2 is not required for NC initiation, maintenance, or progression in *Pdx1*-expressing *Sox2*-absent tissues, supporting a broader conclusion that SOX2 is dispensable for both hijacking and reprogramming mechanisms of BRD4::NUTM1-driven transformation—regardless of the tissue of origin or the SOX2 status of the cell of origin.

### Transcriptomic profiling reveals minimal impact of *Sox2* loss on mNC gene expression programs

To investigate whether *Sox2* loss alters the transcriptional landscape of NC, we performed bulk RNA sequencing on mNC tumors derived from the oral mucosa (KRT14-Cre) and pancreas (Pdx1-Cre) in both *Sox2* WT and *Sox2*-deficient backgrounds. Alignment of sequencing reads confirmed successful *Sox2* deletion in knockout tumors, as visualized in the IGV track (Fig 5A). Principal component analysis revealed that tumors primarily clustered by tissue of origin, rather than by *Sox2* status (Fig 5B), indicating that tissue-specific transcriptional profiles dominate over any effect of SOX2 loss. Consistent with this, differential expression analysis using *DESeq2* identified only a modest number of differentially expressed genes (DEGs) between *Sox2* knockout and WT tumors in each tissue ($|\log_2\text{FC}| > 1$, adjusted $P < 0.05$): 407 up-regulated and 832 down-regulated genes in oral tumors, and 106 up-regulated and 346 down-regulated genes in pancreatic tumors (Fig 5C and D). The labeled genes in the MA plots represent top-ranked examples by statistical significance and fold change, but they span diverse and largely unrelated categories without clear biological relevance to NC. This lack of a coherent or functionally convergent pattern further supports the conclusion that Sox2 loss does not reprogram the core oncogenic transcriptional program. Only a small subset of genes (4 up-regulated, 13 down-regulated) was shared between the two tissue types (Fig 5E). Sox2 appears among the shared down-regulated genes, which serves as an internal control confirming efficient gene deletion in both models, but does not reflect a biologically informative change. Importantly, canonical NC oncogenic drivers, including *Myc*, *Trp63*, and *Brd4::Nutm1*, remained robustly expressed and unaffected by SOX2 status (Fig 5F). To evaluate whether other SOX family members, particularly those within the SOXB clade (*Sox1*, *Sox3*, *Sox14*, *Sox21*), were compensatorily up-regulated, we examined expression levels of all mouse *Sox* genes. No evidence of compensatory up-regulation was observed (Fig 5F). In addition to NC markers and Sox family genes, we

---

**Figure 2. SOX2 is not required for NC initiation and progression in KRT14-expressing tissues.**
**(A)** Schematic of the NCT mouse model with Krt14-Cre–mediated conditional deletion of *Sox2* and activation of *Brd4::Nutm1* fusion in *Krt14*-expressing lineages (oral mucosa, skin, salivary glands). **(B)** Representative bioluminescence images of KRT14-Cre; NCT+/−;*Sox2*fl/fl (*Sox2* KO) and KRT14-Cre; NCT+/− (*Sox2* WT) mice. **(C)** Tumor distribution across anatomical sites in *Sox2* WT, heterozygous KO, and homozygous KO mice. **(D)** Kaplan–Meier survival curves for the three genotypes. *n* = number of independent mice per group (indicated on plot). A log-rank test was used; *P*-values are shown. **(E)** Representative H&E staining of *Sox2* KO and WT tumors, showing poorly differentiated squamous morphology with abrupt keratinization, consistent with NC (n = 3). Scale bar: 800 μm (overview); 50 μm (zoom-in). **(F)** Immunohistochemistry (IHC) for canonical NC markers (BRD4::NUTM1, p63, MYC), Ki-67, and SOX2 in *Sox2* KO and WT tumors (n = 3). Scale bar: 20 μm. **(G)** Quantification of marker-positive tumor cells in *Sox2* WT and KO tumors. Bars indicate the mean ± SD (n = 3). A two-sided unpaired *t* test was used. The significance level was indicated on the plot. *ns*, not significant.

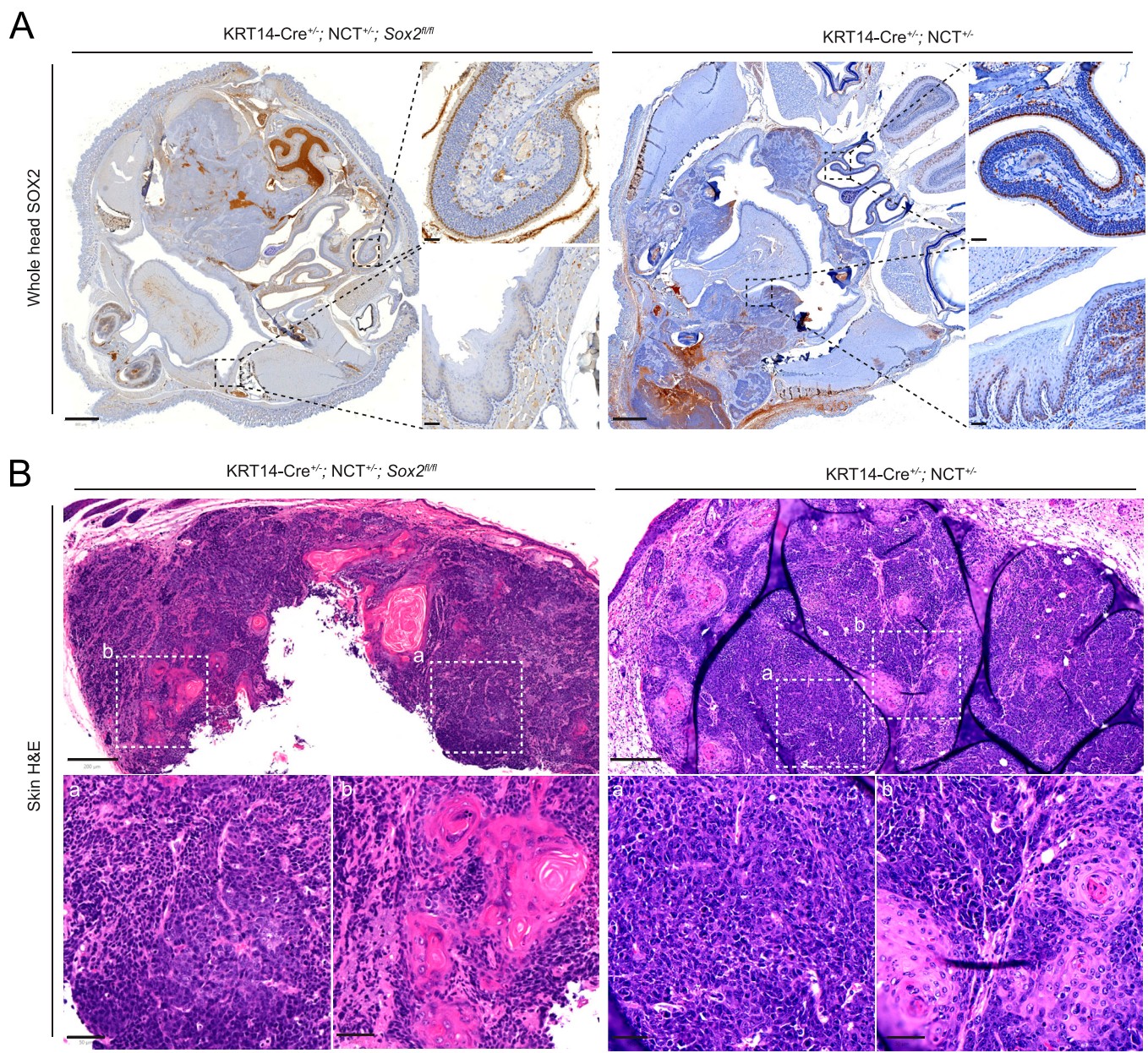

**Figure 3. Effective *Sox2* deletion in target epithelia and SOX2-independent skin tumor formation.**
**(A)** IHC of whole-head sections showing SOX2 expression in oral mucosa, nasal epithelium, and oral tumors of *Sox2* WT and KO mice. **(A)** Scale bar: 800 *µ*m (overview); 50 *µ*m (zoom-in). *Note: The SOX2 image in* (A) *(bottom right subpanel) and the SOX2 panel in* Fig 1B *(first row) were derived from the same whole-slide scan of the same tissue section, shown at different magnifications and regions of interest.* **(A, B)** Representative H&E images of skin tumors from *Sox2* KO and WT mice revealing poorly differentiated squamous carcinoma (A) with abrupt keratinization (B). Scale bar: 200 *µ*m (overview); 50 *µ*m (zoom-in).

assessed expression of candidate SOX2 partner factors, including *Pou5f1*, *Nanog*, *Klf4*, *Klf5*, *Tead1–4*, *Yap1*, and *Wwtr1*. None of these genes were differentially expressed between Sox2 WT and knockout tumors in either oral or pancreatic NC (Fig 5F). To further explore whether *Sox2* loss subtly reprograms the transcriptome, we performed gene set enrichment analysis (GSEA) using the full ranked gene list (based on shrunken $\log_2$ fold changes). Across both tissues, only a limited number of gene sets were significantly down-regulated in the Sox2-deficient tumors. Notably, the commonly suppressed pathways included oxidative phosphorylation, proton motive force–driven mitochondrial ATP synthesis, cellular respiration, aerobic respiration, and purine nucleoside/ribonucleoside triphosphate biosynthesis, suggesting a modest reduction in metabolic and biosynthetic activity (Fig 5G).

Together, these findings indicate that SOX2 loss does not lead to major transcriptional rewiring in *Brd4::Nutm1*-driven tumors. Aside from a mild impact on mitochondrial and biosynthetic gene expression, the core oncogenic transcriptional program remains intact, reinforcing the conclusion that SOX2 is dispensable for NC maintenance at both the molecular and phenotypic levels.

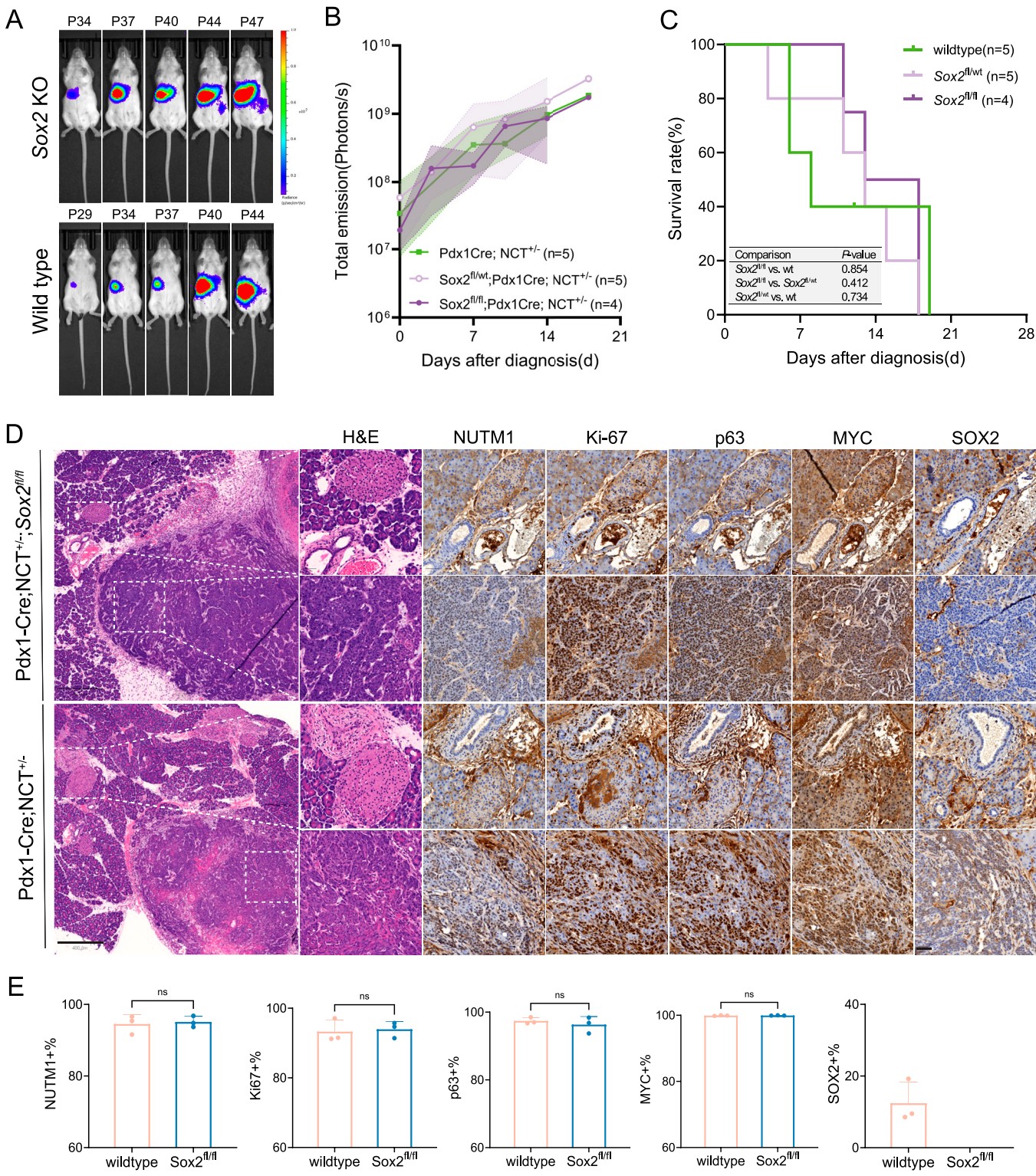

**Figure 4. SOX2 is dispensable for NC initiation and progression in SOX2-negative lineages.**
**(A)** Representative bioluminescence images of Pdx1-Cre; NCT[+/−];Sox2[fl/fl] (*Sox2* KO) and Pdx1-Cre; NCT[+/−] (*Sox2* WT) mice. **(B)** Quantification of bioluminescence signal (photons/s) over time. Lines represent mean values; shaded areas indicate the range. **(C)** Kaplan–Meier survival curves for *Sox2* WT, heterozygous KO, and homozygous KO mice. *n* = number of independent mice per group (indicated on plot). A log-rank test was used; *P*-values are shown. **(D)** Representative H&E staining and IHC for canonical NC markers (BRD4::NUTM1, p63, MYC), Ki-67, and SOX2 (n = 3). Scale bar: 800 $\mu$m (overview); 50 $\mu$m (zoom-in). **(E)** Quantification of marker-positive tumor cells in *Sox2* WT and KO tumors. Bars indicate the mean ± SD (n = 3). A two-sided unpaired *t* test was used. The significance level was indicated on the plot. *ns*, not significant.

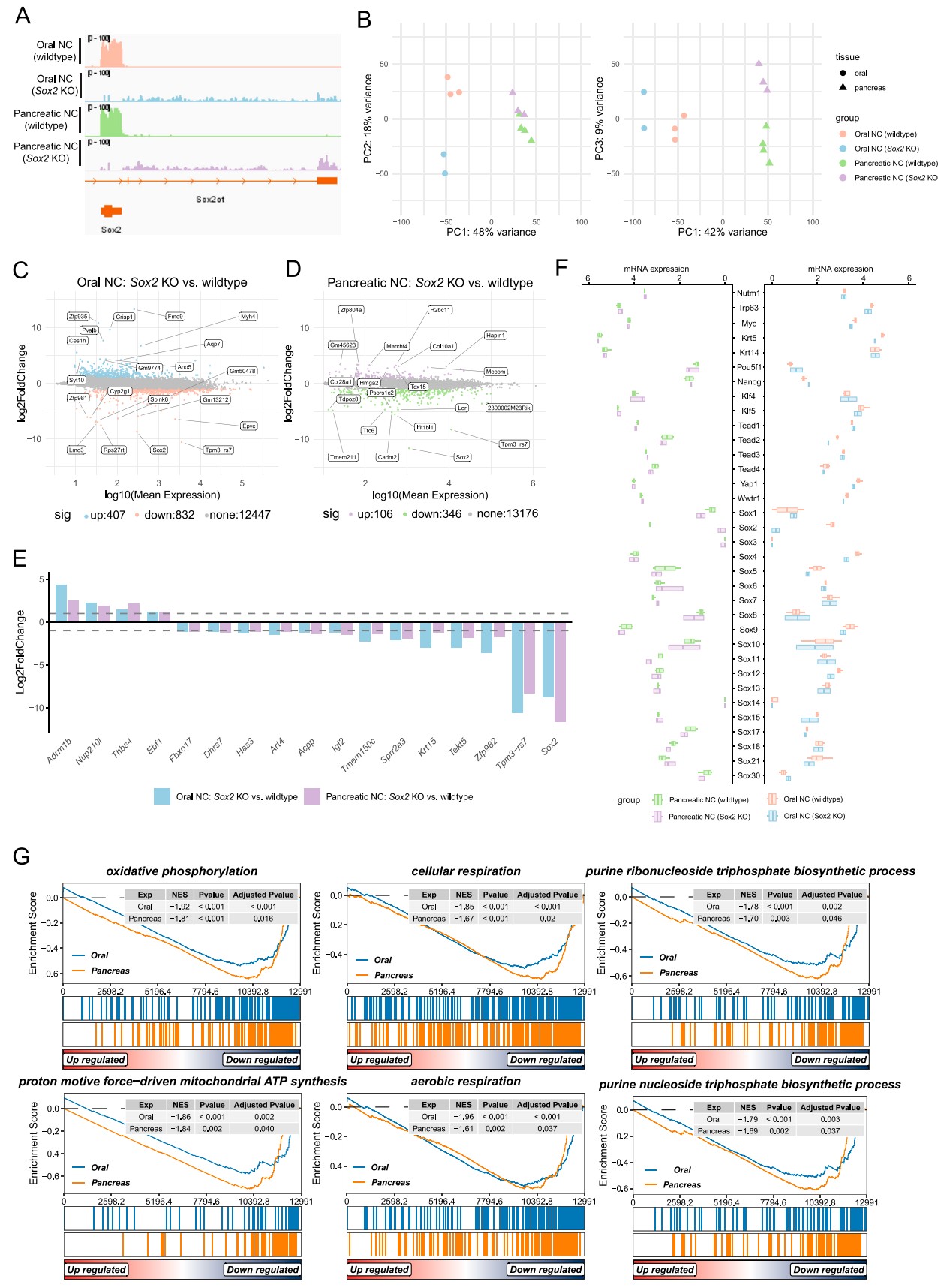

# Discussion

NUT carcinoma (NC) is an aggressive and poorly differentiated squamous cell cancer driven by *BRD4::NUTM1* and other *NUTM1* fusion oncogenes. Despite their diverse anatomical presentation, NCs consistently exhibit a squamous cell phenotype characterized by basal cell marker expression and high proliferative capacity (3, 5, 7). This phenotypic uniformity across tissues suggests that BRD4::NUTM1 may either hijack existing squamous cell programs or actively reprogram diverse progenitor cells toward a shared on-cogenic state. Among potential cooperating factors, SOX2 has been proposed as a critical contributor to NC biology, as a master regulator of stemness and squamous fate.

SOX2 has been implicated in the pathogenesis of various cancers, often reflecting the biology of the cell of origin. In tumors arising from SOX2$^+$ epithelial progenitor populations, such as those in the upper aerodigestive tract, SOX2 is frequently amplified (in the 3q chromosomal region) or overexpressed and contributes to maintaining the proliferative, undifferentiated SCC state (32, 33, 34, 35, 36). In these contexts, SOX2 often functions as a lineage-survival oncogene, reinforcing the developmental identity of the tissue and cooperating with other oncogenic signals such as TP63 and PI3K pathway activation, and is often insufficient alone to drive transformation (36, 37, 38, 39). In skin SCC specifically, SOX2 is not expressed in the interfollicular epidermis under homeostatic conditions, but marks tumor-initiating cells in mouse models of skin SCC and is required for maintaining their self-renewal and tumorigenic capacity (18). In contrast, tumors that arise from SOX2-negative lineages ectopically activate SOX2 during tumor pro-gression, particularly in association with squamous transdifferentiation, but its functional contribution in these contexts appears variable or non-essential (40, 41, 42, 43, 44, 45, 46, 47). In parallel to carcinoma, SOX2 is a critical oncogene for neural progenitor cell–derived cancers, such as glioblastoma and medulloblastoma, where it sustains cancer stem cells (48, 49). These observations highlight the importance of developmental context in interpreting SOX2 function in cancer and suggest that its contribution to tumorigenesis may be tightly linked to its physiological role in tissue-specific progenitor populations and conditioned to specific oncogenic mechanism.

In line with this variability, Stirnweiss et al compared drug response profiles and genetic features of 12 NC cell lines and identified BETi-sensitive and BETi-insensitive subgroups (27). Notably, SOX2 was not among the DEGs distinguishing these groups, suggesting it is not a key determinant of therapy response. Together with the high variability of SOX2 expression observed in these 12 NC cell lines (Fig S1), these findings support the conclusion that SOX2 is not universally required for NC maintenance or therapy sensitivity.

Here, we use multiple lineage-specific GEMMs and tran-scriptomic analyses to demonstrate that SOX2 is dispensable for BRD4::NUTM1-driven NC initiation, maintenance, and progression, regardless of tissue of origin. In SOX2-expressing tissues such as the oral mucosa and salivary glands, deletion of *Sox2* in KRT14-Cre–driven models did not impair tumor formation, growth ki-netics, or histological identity. Tumors retained canonical NC features, including the expression of BRD4::NUTM1, P63, and MYC, and showed no reduction in proliferation. Importantly, the ability of BRD4::NUTM1 to drive NC from SOX2-negative lineages, such as skin and pancreas, was unaffected by the absence of SOX2, indicating that even when BRD4::NUTM1 may rely on a reprog-ramming mechanism (rather than hijacking preexisting SOX2-driven circuits), SOX2 is not required. Bulk RNA-seq further supported these findings by showing minimal transcriptomic differences between *Sox2* WT and knockout tumors. Principal component analysis revealed clustering primarily by tissue type rather than genotype, and differential gene expression analysis identified only modest changes, with very few shared DEGs across tissues. Importantly, the expression of other SOX family members did not increase, ruling out compensatory redundancy. GSEA revealed slight down-regulation of oxidative phosphorylation and purine metabolism pathways in Sox2-deficient tumors, suggesting a mild reduction in biosynthetic and mitochondrial activity.

Collectively, these findings challenge the assumption that SOX2 is a universal driver or necessary cofactor in NC oncogenesis. Although SOX2 is expressed in subpopulations of both human and mouse NC cells, its presence appears essential neither for tumor initiation nor for maintaining oncogenic transcriptional programs once NC is established. This distinction is particularly important given prior reports linking SOX2 to pluripotency, squamous lineage fidelity, and tumor-initiating potential. Our data suggest that in the context of BRD4::NUTM1 fusion, SOX2 is neither a gatekeeper nor a central node in the core transcriptional circuitry of NC.

From a clinical perspective, these results carry implications for therapeutic targeting. Although SOX2 is a biomarker of interest in various squamous cell cancers and has been proposed as a valuable therapeutic target (15, 50), our study suggests that SOX2-directed therapies are unlikely to be effective in NC, especially as monotherapy. Instead, efforts should focus on directly targeting the fusion oncoprotein or downstream transcriptional and chro-matin remodeling dependencies.

In conclusion, our findings demonstrate that SOX2 is not re-quired for BRD4::NUTM1-mediated NC, regardless of the tissue lineage or the developmental context of the cell of origin, at least in mouse models. This underscores the remarkable autonomy of the BRD4::NUTM1 fusion in establishing and maintaining the NC phenotype, challenging prior assumptions about the universal role of SOX2 in NC oncogenesis. Moving forward, it will be critical to

---

**Figure 5. Transcriptomic profiling reveals minimal impact of *Sox2* loss on mNC gene expression programs.**
**(A)** Integrative Genomics Viewer (IGV) tracks showing RNA-seq read coverage over the *Sox2* locus in *Sox2* WT and KO tumors from oral and pancreas. **(B)** Principal component analysis of *Sox2* WT and KO tumors from oral and pancreas. n = 4 (oral WT), 2 (oral KO), 4 (pancreatic WT), 3 (pancreatic KO). **(C, D)** MA plots displaying differentially expressed genes (DEGs) in *Sox2* KO versus WT tumors from oral mucosa (C) and pancreas (D). DEGs were defined as |log$_2$FC| > 1, adjusted *P* < 0.05. **(E)** Grouped bar graph showing log$_2$ fold changes of DEGs overlapping between oral and pancreatic tumors. **(F)** Bar plots showing expression levels (log$_{10}$(normalized count + 1), normalized using the DESeq2 mean-of-ratios method) of *Brd4::Nutm1*, *Trp63*, *Myc*, candidate *Sox2* partners, and *Sox* family genes in *Sox2* WT and KO tumors. **(G)** Gene set enrichment analysis of *Sox2* KO versus WT tumors from oral and pancreatic tissues.

identify the core transcriptional and chromatin remodeling dependencies that sustain NC across contexts, and to investigate whether rare SOX2-positive subpopulations contribute to tumor heterogeneity, therapeutic resistance, or recurrence. Finally, we acknowledge several limitations of this study. Our conclusions are based on in vivo mouse models, and species-specific differences in SOX2 function may exist. Although human NC cell studies have shown that SOX2 knockdown reduces colony and tumorsphere formation but has limited effects on proliferation [20], our results indicate that SOX2 is dispensable for tumor initiation and maintenance in mice. In addition, although we examined candidate SOX2 partner genes and found no changes upon *Sox2* loss, expression data alone cannot confirm functional interactions. Future studies using human-based models and direct assays of SOX2 partner binding will be important to define potential context-dependent roles. Taken together, our results underscore the autonomy of BRD4::NUTM1 in sustaining the NC phenotype and sharpen the focus on BRD4::NUTM1-directed therapeutic strategies.

# Materials and Methods

### Ethical statement

All animal work was carried out under PROTO 202300127, approved by the Michigan State University (MSU) Campus Animal Resources (CAR) and Institutional Animal Care and Use Committee (IACUC) in AAALAC-credited facilities.

### Mouse lines and breedings

The NUT carcinoma translocator (NCT) model (MMRRC_071753-MU) was generated as previously described (14). Lineage-specific Cre driver lines were used to induce the *Brd4::Nutm1*-forming t(2; 17) chromosome translocation in defined tissues: KRT14-Cre (JAX 018964) for oral mucosa, skin, and salivary gland (29), Nkx2.1-Cre (JAX 008661) for lung and thyroid NC (51), and Pdx1-Cre (JAX 014647) for pancreatic NC (52). To generate experimental cohorts, NCT mice were crossed with these Cre lines to produce offspring harboring both the *Brd4-loxp and Nutm1-loxp* alleles and the appropriate Cre transgene. For conditional deletion of *Sox2*, *Sox2*$^{fl/fl}$ mice (JAX 013093) were used (53). *Sox2*$^{fl/fl}$ mice were first crossed with NCT mice to generate NCT$^{+/+}$; *Sox2*$^{fl/fl}$ animals. In parallel, *Sox2*$^{fl/fl}$ mice were bred with lineage-specific Cre drivers (KRT14-Cre or Pdx1-Cre) to generate homozygous Cre$^+$; *Sox2*$^{fl/fl}$ mice. These two homozygous lines were then intercrossed to produce offspring carrying the *Brd4::Nutm1* allele, lineage-specific Cre, and biallelic *Sox2* floxed alleles, enabling complete deletion of *Sox2* in the target tissue. Genotyping was performed on ear or tail biopsies using PCR with primers specific for the *Brd4::Nutm1* fusion allele, Cre transgenes, and floxed *Sox2* alleles according to standard protocols. Primer sequences are listed in Table S1.

### In vivo bioluminescence imaging

Mice were injected with D-luciferin sodium solution intraperitoneally (150 mg per kilogram body weight, 15 mg/ml in PBS, GoldBio) 15–20 min before imaging. Bioluminescent images were acquired in both dorsal and ventral positions using the IVIS Spectrum In Vivo Imaging System (PerkinElmer) under isoflurane anesthesia. The total flux (photons/second) within the regions of interest was quantified using Living Image software (PerkinElmer). The mean of the total flux from both dorsal and ventral images was calculated for each mouse and used for generating tumor growth curves.

### Tissue preparation and immunohistochemistry

Tissue samples were harvested and fixed in 4% PFA at 4°C for 24 h. Mouse heads were decalcified in 10% EDTA at 4°C for 12–14 d, with the solution replaced every 2 d. Paraffin-embedded tissues were sectioned at 5 $\mu$m in serial and stained with hematoxylin–eosin (H&E) or subjected to immunohistochemistry (IHC) using standard procedures. The following primary antibodies were used for IHC analysis: NUTM1 (1:100, HA721690; Huabio), Ki-67 (1: 800, 12202; Cell Signaling Technology), c-MYC (1:500, 10828-1-AP; Proteintech), SOX2 (1:1,000, AF2018; R&D Systems), p63 (1:1,000, ab124762; Abcam). Histopathological evaluation was performed by a board-certified veterinary pathologist (MFT).

### Histological quantification

Whole-slide scans were generated using the Zeiss Axioscan 7 or the Aperio VERSA Slide Scanner at 20x. Quantitative analysis of immunoreactivity was conducted using the open-source QuPath software (v0.5.1) (54). For each marker, the "Positive Cell Detection" tool was applied to tumor regions, with manual optimization of parameters including pixel size (optical density sum) and nucleus DAB optical density mean threshold to minimize false-positive detection. All other settings were left at default values. The percentage of marker-positive tumor cells was recorded for each sample.

### Bulk RNA-seq

WT and *Sox2* knockout (KO) tumors were dissected and immediately submerged in RNAlater (Thermo Fisher Scientific) at 4°C for 24 h, followed by storage at –80°C until library preparation. Total RNA was extracted using RNeasy Mini Kit (QIAGEN) according to the manufacturer's protocol. One microgram of RNA per sample was subjected to ribosomal RNA (rRNA) depletion and library preparation using the Illumina Stranded Total RNA Prep with Ribo-Zero Plus kit, following the manufacturer's instructions. Library concentration and size distribution were assessed using the Agilent 4200 TapeStation. Sequencing was performed by St. Jude Children's Research Hospital, generating ~50 million paired-end reads per sample.

## Gene expression analysis

Raw reads were trimmed with Trim Galore (v0.6.10) with the default parameters. FastQC (v0.12.1) was used for data quality assessment. Trimmed reads were quantified with Kallisto (v0.50.0) (55) using the GENCODE mouse transcriptome (GRCm38, release M23) with settings *--rf-stranded -b 100*. Transcript-level quantifications were imported into R (v4.3.2) and summarized to gene-level counts using the *tximport* package (v1.28.0) (56). Differential gene expression analysis was conducted using *DESeq2* (57), applying the Wald tests. Genes were filtered to retain those with ≥10 reads in at least two samples. Log$_2$ fold changes (LFCs) were shrunk using the "apeglm" method to improve effect size estimation (58). DEGs were defined as those with |LFC| > 1 and adjusted $P$ < 0.05 (Benjamini–Hochberg correction (59)). GSEA (60) was performed using *clusterProfiler* (v4.8.2) (61), testing against the MSigDB (Hallmark) (62), Gene Ontology (GO) (63, 64), Reactome (65), and KEGG (66) databases. Genes were ranked by shrunken LFC, and pathways with an adjusted $P$ < 0.05 were considered statistically significantly enriched.

All sequencing data have been deposited in the NCBI Gene Expression Omnibus (GEO) under the accession number GSE300412.

## Statistics and reproducibility

Data are presented as means ± SD or median (interquartile range, IQR) and represent a minimum of three independent experiments. Statistical parameters including statistical analysis, statistical significance, and *n*-values are described in the figure legends. Statistical analyses were carried out using Prism 10 Software (GraphPad). Statistical comparisons of two groups were performed using the two-sided unpaired *t* test, and comparisons of more than three groups were performed by one-way analysis of variance (ANOVA). Cumulative rates of survival were calculated by the Kaplan–Meier method and compared between groups by the log-rank test. A value of $P$ < 0.05 was considered statistically significant. The presented data are a representative image of more than three biologically independent experiments.

# Data Availability

The raw and processed RNA-sequencing data have been deposited to Gene Expression Omnibus (GSE300412). All the remaining data are available from the corresponding author upon request.

# Supplementary Information

# Acknowledgements

We thank Dr. Ehab Hanna from the MD Anderson Cancer Center and Dr. Diana Bell from the University of Pittsburgh Medical Center for constructive discussions. We thank MSU Histology Laboratory at Veterinary Diagnostic Laboratory (Taylor Vaughn) for their help in histopathology and IHC analysis and the Q-BEAM facility for assistance in slide scanning. We thank the Advanced Technology and Genomics (ATG) Core at the St. Jude Research for assistance in bulk RNA sequencing. This research was supported by an R37 grant (R37CA269076) and an R03 grant (R03CA286646) from the National Institute of Health (NIH)/National Cancer Institute (NCI), both awarded to B Gu.

## Author Contributions

C Luo: conceptualization, data curation, formal analysis, validation, investigation, visualization, methodology, and writing—review and editing.
D Zheng: formal analysis, validation, investigation, visualization, methodology, and writing—review and editing.
A Elnegiry: data curation, formal analysis, investigation, and methodology.
S Bhatara: software and methodology.
GI Mias: methodology and writing—review and editing.
MF Tsoi: data curation, formal analysis, visualization, methodology, and writing—review and editing.
J Yu: methodology and writing—review and editing.
B Gu: conceptualization, data curation, formal analysis, supervision, funding acquisition, validation, investigation, visualization, methodology, project administration, and writing—original draft, review, and editing.

## Conflict of Interest Statement

The authors declare that they have no conflict of interest.

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
