## [Reviewer comments · Life Science Alliance]

SOX2 is a dispensable modulator of NUT carcinoma oncogenesis in mice

Chenxiang Luo, Dejin Zheng, Ahmed Elnegiry, Sheetal Bhatara, George Mias, Mayra Tsoi, Jiyang Yu, and Bin Gu
DOI: <https://doi.org/10.26508/lsa.202503447>

Corresponding author(s): Bin Gu, Michigan State University

Review Timeline:

Submission Date:	2025-07-08
Editorial Decision:	2025-08-27
Revision Received:	2025-10-08
Editorial Decision:	2025-11-04
Revision Received:	2025-11-12
Accepted:	2025-11-13

Scientific Editor: Sarita Hebbar

Transaction Report:

August 27, 2025

Re: Life Science Alliance manuscript #LSA-2025-03447

Dr. Bin Gu
Michigan State University
Obstetrics, Gynecology & Reproductive Science
3319 Bioengineering Building 775 Woodlot Dr.
East Lansing, MI 48824

Dear Dr. Gu,

Thank you for submitting your manuscript entitled "SOX2 is a dispensable modulator of NUT carcinoma oncogenesis in mouse" to Life Science Alliance. The manuscript was assessed by three expert reviewers, whose comments are appended to this letter.

As you will note all three reviewers found your work on this murine NUT carcinoma model of value to the community. That said, they have raised several important concerns with the results and discussion section that must be addressed before publication.

- Reviewers 1 and 3 had important concerns on the experimental data. We agree with the Reviewer 3 that the number of mice in the survival studies (Fig 2D and 4C) must be increased. We also agree with Reviewer 1 that quantification of immunohistochemistry data pertaining to Figure 4D must be included. Further we recommend you follow the suggestion of Reviewer 3 to rule out Sox 2 staining in tumour cells with double labelling (Sox 2 and fusion driver)
- Reviewer 2 makes an important point on the consequence of Sox2 knockdown in human cell lines. We leave it to you to follow up on this in an experimental model of your choice, or address this limitation in the discussion. Likewise, we agree with Reviewer 2 that the mechanism underlying Sox2's function in this model is missing. We understand that additional experiments in this regard may be difficult in which case we expect that you elaborate on these points in the discussion.
- Finally, we urge you to factor in the relevant points raised by Reviewer 1 and Reviewer 2 on contextualising the therapeutic aspect of these findings in an appropriate manner.

We invite you to submit a revised manuscript addressing the Reviewer comments. When submitting the revision, please include a letter addressing the reviewers' comments point by point. While a rebuttal must respond to all points in some form, additional experiments to resolve these points, other than indicated above, will not be required.

Thank you for this interesting contribution to Life Science Alliance. We are looking forward to receiving your revised manuscript.

Sincerely,

Sarita Hebbar, PhD
Scientific Editor
Life Science Alliance
<http://www.lsajournal.org>

- A letter addressing the reviewers' comments point by point.
- An editable version of the final text (.DOC or .DOCX) is needed for copyediting (no PDFs).

B. MANUSCRIPT ORGANIZATION AND FORMATTING:

Reviewer #1 (Comments to the Authors (Required)):

Summary:

Here, Luo and colleagues present a straightforward series of results showing that SOX2 is dispensable for the initiation or maintenance of NUT carcinoma in a murine model. Because SOX2 has previously been implicated in NUT carcinoma, this new study is significant, particularly because the work here uses an animal model as opposed to the in vitro model used previously. The current manuscript is comprehensive, investigating the role of SOX2 in SOX2+ and SOX2- lineages and incorporates molecular analysis at the level of RNA-seq. The only caveat, which the authors already acknowledge, is whether there a species-specific difference such that the results are specific to murine NUT carcinoma. However, resolving this point is outside the scope of the current manuscript. With minor revisions, I think this manuscript will be suitable for publication.

Major comments:

A portion of the abstract and discussion interprets the findings here in the context of therapeutic intervention. However, to my knowledge, there are no therapies targeting SOX2 and SOX2 has not been prioritized as a target in NUT carcinoma. BET inhibitors targeting BRD4-NUT (and other BRDs) have been prioritized as a targeted therapy in NUT carcinoma. Therefore, I think these portions of the manuscript should be revisited with respect to their framing.

Minor comments:

Fig. 2G - The authors should show the individual datapoints for each tumor overlaid on the bar charts. Therefore, the reader would better understand the distribution of the datapoints.

Fig. 4D - Can the authors please quantify the percent of cells expressing each marker as in Fig. 2G?

Fig. 5C and 5D - Is there any significance to the labeled genes in the MA plots?

Fig. 5E - It is not clear why the authors consider Sox2 as a shared downregulated gene between the two tissue types. The authors have clearly shown that Sox2 has been deleted, so the significance of Sox2 in this panel is unclear and even unnecessary.

Reviewer #2 (Comments to the Authors (Required)):

SOX2 has been shown to be upregulated BRD4-NUT NUT-Midline carcinomas in patients and SOX2 expression has been identified in human cell lines, but little functional work has been performed on the role of SOX2 in NUT-Midline carcinogenesis. Luo et al provide an important and rigorous study of negative data in a mouse model of NUT-Midline Carcinoma show it is dispensable for carcinogenesis. SOX2 expression is heterogenous in patients and is not uniformly expressed within tumors of

both mice and humans. SOX2 knockout shows little effect on initiation, proliferation, or gene expression in their model. This reviewer is satisfied with the data presented, but it is of limited scope and potential utility. I would suggest expanding these data outside of the one mouse model. Utilizing the one mouse model is the major caveat to these data, and any orthogonal investigation by modulating SOX2 in another system, and preferably a human model, would be greatly appreciated. It would strengthen these conclusions and the potential translation. I would suggest a revision to include some of these studies.

Major points:

Little work has been done in human models of NUT-Midline carcinoma. Does SOX2 knockdown provide a phenotype? Is SOX2 knockdown dispensable for ongoing proliferation of cell lines? Simple siRNA experiments in a handful of cell lines would greatly substantiate the authors claims in a human model. This can be done in a short timeframe, on the order of a few months.

Does SOX2 expression provide an advantage to tumors in patients? Does it correlate with metastasis, progression, or resistance to therapy in patients? I understand it's hard to determine a lot of this, given the aggressive nature of the disease and its relative rareness, but the authors should try to address this, again either in their animal model or in human cell lines. Knockdown and treatment studies could be easily performed at the same time in a few months. These studies would show a role for SOX2 in NUT-Midline carcinoma that is potentially not dispensable.

It is known that SOX2 cannot function without a heterodimeric partner (<https://doi.org/10.1242/dev.200547>) to cooperate with to specifically engage DNA. Does this model recapitulate expression of a putative partner that could be found in human patients? What are the potential binding partners of SOX2 in patient samples? If there are few, maybe this is why SOX2 expression may be aberrantly upregulated, but lack of a binding partner may render it non-functional. It has been suggested that TP63 may act as a SOX2 partner, but it may not be the partner utilized in these tumors or in this particular genetically engineered mouse model. This would greatly enhance relevance of these data. The initial investigation should be easy to complete in a short time frame, but I understand that the investigation is open ended. If a clear putative partner is found, a co-IP of SOX2 and identification of co-binding could be performed swiftly, and better yet, a mass-spectroscopic analysis of SOX2-bound proteins would be quickly, albeit more expensive. Another approach could be ChIP-seq of SOX2 and motif scanning, which I again understand is technically difficult, expensive, and time consuming. Regardless, it is a clear caveat to these studies if SOX2 is studied in a vacuum without addressing what partnering factor, if any, it is using to engage with DNA.

Reviewer #3 (Comments to the Authors (Required)):

In this study, Luo et al analyzed the importance of Sox2 expression in NUT carcinoma. Using their recently developed GEMM of BRD4::NUTM1-driven NUT carcinoma, they delete Sox2 using a Cre-mediated approach and observe no significant differences in survival or transcriptional programs. While the experiments use an elegant design, there are concerns about the conclusions. The number of mice used in the survival experiments are too low to come to any significant conclusions. Additional mice should be added to increase statistical power.

- Figure 2D&4C: The number of mice used in the survival experiments are too low to come to any significant conclusions (currently $n = 5$ & 6). Each arm should include at least 15-20 mice to allow for any meaningful significant analysis of results. Further, there is a non-significant trend towards longer tumor latency in Sox2 fl/fl versus Sox2 wt mice ($p = 0.085$). If the authors would add additional mice to the study this trend would likely become significant. The median survival is also longer in Sox2 fl/fl mice (21 days versus 14 days). Although the authors mentioned that the growth location of the tumors often impairs mice from drinking, feeding, and breathing and makes it hard to conduct actual "survival" studies, I still find the conclusion that Sox2 loss does not affect tumor growth misleading. The survival graph of pancreatic tumors shows a similar result, although, again the number of mice used is too low and will need to be increased in order to come to a significant conclusion. But also here there is a trend towards longer latencies (6 versus 11 days) and median survival times (8 versus 12 days) in Sox2 fl/fl mice.

- Figure 2F: There is quite a lot of residual SOX2 staining in Sox2 KO mice. Can the authors perform co-IF stainings for SOX2 and the fusion driver to show that the residual SOX2 staining is not in the tumor cells?

Response to Reviewers

Thank you for the opportunity to revise our manuscript entitled "*SOX2 is a dispensable modulator of NUT carcinoma oncogenesis in mouse*". We sincerely appreciate the reviewers' constructive feedback and have revised our manuscript per these comments. Our point-by-point responses are listed below. All textual revisions are highlighted in the revised manuscript.

Reviewer #1 (Comments to the Authors (Required)):

Summary:

Here, Luo and colleagues present a straightforward series of results showing that SOX2 is dispensable for the initiation or maintenance of NUT carcinoma in a murine model. Because SOX2 has previously been implicated in NUT carcinoma, this new study is significant, particularly because the work here uses an animal model as opposed to the in vitro model used previously. The current manuscript is comprehensive, investigating the role of SOX2 in SOX2+ and SOX2- lineages and incorporates molecular analysis at the level of RNA-seq. The only caveat, which the authors already acknowledge, is whether there is a species-specific difference such that the results are specific to murine NUT carcinoma. However, resolving this point is outside the scope of the current manuscript. With minor revisions, I think this manuscript will be suitable for publication.

We are delighted by the interest and comments of the Reviewer. Thank you for appreciating our work and for providing thoughtful suggestions and feedback.

Major comments:

A portion of the abstract and discussion interprets the findings here in the context of therapeutic intervention. However, to my knowledge, there are no therapies targeting SOX2 and SOX2 has not been prioritized as a target in NUT carcinoma. BET inhibitors targeting BRD4-NUT (and other BRDs) have been prioritized as a targeted therapy in NUT carcinoma. Therefore, I think these portions of the manuscript should be revisited with respect to their framing.

We thank the reviewer for this important point. We have revised the abstract and discussion to avoid overemphasis on SOX2 as a therapeutic target (line 38-40 and line 430-432).

Minor comments:

Fig. 2G - The authors should show the individual datapoints for each tumor overlaid on the bar charts. Therefore, the reader would better understand the distribution of the datapoints.

We have revised Fig. 2G to overlay individual datapoints on the bar charts, enabling visualization of data distribution.

Fig. 4D - Can the authors please quantify the percent of cells expressing each marker as in Fig. 2G?

We have now quantified the percentage of marker-positive cells in Fig. 4D, consistent with the methodology in Fig. 2G. These quantifications are included in the revised figure (revised Fig. 4E).

Fig. 5C and 5D - Is there any significance to the labeled genes in the MA plots?

We thank the reviewer for raising this point. The labeled genes in Fig. 5C/D were selected simply as representative examples based on statistical significance and fold change. These include diverse metabolic enzymes, extracellular matrix components, neuronal regulators, and uncharacterized loci. Importantly, none are established Sox2 targets or known to be central to NUT carcinoma biology. Their lack of clear relevance highlights that the observed transcriptional differences are scattered and do not point to a coherent oncogenic program, further supporting our conclusion that Sox2 loss does not reprogram the core transcriptional circuitry of NC. We have clarified this point in the revised Results section (line 314-318).

Fig. 5E - It is not clear why the authors consider Sox2 as a shared downregulated gene between the two tissue types. The authors have clearly shown that Sox2 has been deleted, so the significance of Sox2 in this panel is unclear and even unnecessary.

We thank the reviewer for this comment. We agree that Sox2 downregulation is an expected outcome of our genetic deletion strategy and does not provide new biological insight. However, we have retained Sox2 in Fig. 5E as an internal control to illustrate the effectiveness of Sox2 deletion across both tumor models. In the revised Results section (line 320-322), we have clarified that Sox2 is shown only to confirm successful gene knockout, and not interpreted for biological significance.

Reviewer #2 (Comments to the Authors (Required)):

SOX2 has been shown to be upregulated BRD4-NUT NUT-Midline carcinomas in patients and SOX2

expression has been identified in human cell lines, but little functional work has been performed on the role of SOX2 in NUT-Midline carcinogenesis. Luo et al provide an important and rigorous study of negative data in a mouse model of NUT-Midline Carcinoma show it is dispensable for carcinogenesis. SOX2 expression is heterogenous in patients and is not uniformly expressed within tumors of both mice and humans. SOX2 knockout shows little effect on initiation, proliferation, or gene expression in their model. This reviewer is satisfied with the data presented, but it is of limited scope and potential utility. I would suggest expanding these data outside of the one mouse model. Utilizing the one mouse model is the major caveat to these data, and any orthogonal investigation by modulating SOX2 in another system, and preferably a human model, would be greatly appreciated. It would strengthen these conclusions and the potential translation. I would suggest a revision to include some of these studies.

We thank the Reviewer for positively evaluating our work and for providing insightful comments.

Major points:

Little work has been done in human models of NUT-Midline carcinoma. Does SOX2 knockdown provide a phenotype? Is SOX2 knockdown dispensable for ongoing proliferation of cell lines? Simple siRNA experiments in a handful of cell lines would greatly substantiate the authors claims in a human model. This can be done in a short timeframe, on the order of a few months.

We thank the reviewer for this valuable suggestion. Prior work in human NC cell lines have addressed this question, showing that SOX2 knockdown suppresses colony and tumorsphere formation while only modestly affecting proliferation (Wang et al., *Cancer Res* 2014). We had described these findings in the Introduction (lines 90–92), but in response to the reviewer's comment we have now explicitly emphasized them again in the Discussion (line 424-430). Specifically, we revised the closing section of the Discussion to contrast these published cell line findings with our in vivo GEMM data, which demonstrate that SOX2 loss does not impair tumor initiation, growth, or maintenance. This addition clarifies how our study extends prior work and supports the conclusion that SOX2 is dispensable for NC oncogenesis.

Does SOX2 expression provide an advantage to tumors in patients? Does it correlate with metastasis, progression, or resistance to therapy in patients? I understand it's hard to determine a lot of this, given the aggressive nature of the disease and its relative rareness, but the authors should try to address this, again either in their animal model or in human cell lines. Knockdown and treatment studies could be easily performed at the same time in a few months. These studies

would show a role for SOX2 in NUT-Midline carcinoma that is potentially not dispensable.

We thank the reviewer for this important point. Clinical correlation data for SOX2 expression in NC remain scarce (Results, lines 206–215), and most available cell lines are derived from advanced, heavily treated tumors. Nevertheless, an existing study provides useful insights. Stirnweiss et al. (*Oncotarget*, 2017) compared drug response profiles and genetic features of 12 NC cell lines, identifying BET inhibitor–sensitive and –insensitive subgroups. Notably, SOX2 was not among the differentially expressed genes that distinguished these groups, suggesting it is not a key determinant of therapy response. Consistent with this, our analysis of these 12 NC cell lines (Fig. 1D) showed highly variable SOX2 expression, whereas BRD4::NUTM1 expression was consistent. Together with our GEMM data showing that SOX2 loss does not impair tumor initiation, growth, or maintenance, these findings support the conclusion that SOX2 is not required for NC oncogenesis. We have now expanded the Discussion to incorporate these points and to highlight the need for future human-based studies to clarify potential context-dependent roles of SOX2 (line 374-380 and line 422-432).

It is known that SOX2 cannot function without a heterodimeric partner (<https://doi.org/10.1242/dev.200547>) to cooperate with to specifically engage DNA. Does this model recapitulate expression of a putative partner that could be found in human patients? What are the potential binding partners of SOX2 in patient samples? If there are few, maybe this is why SOX2 expression may be aberrantly upregulated, but lack of a binding partner may render it non-functional. It has been suggested that TP63 may act as a SOX2 partner, but it may not be the partner utilized in these tumors or in this particular genetically engineered mouse model. This would greatly enhance relevance of these data. The initial investigation should be easy to complete in a short time frame, but I understand that the investigation is open ended. If a clear putative partner is found, a co-IP of SOX2 and identification of co-binding could be performed swiftly, and better yet, a mass-spectroscopic analysis of SOX2-bound proteins would be quickly, albeit more expensive. Another approach could be ChIP-seq of SOX2 and motif scanning, which I again understand is technically difficult, expensive, and time consuming. Regardless, it is a clear caveat to these studies if SOX2 is studied in a vacuum without addressing what partnering factor, if any, it is using to engage with DNA.

We thank the reviewer for this important point. To address this, we extended our transcriptomic analysis to include reported candidate SOX2 partners. Specifically, we examined pluripotency factors (*Pou5f1*, *Nanog*, *Klf4*), squamous cofactors (*Klf5*, *Tead1–4*), and Hippo pathway co-activators (*Yap1*, *Wwtr1*). As shown in the revised Fig. 5F, these candidate partners remained unchanged between Sox2 wild-type and knockout tumors in both oral and pancreatic NC models.

Importantly, canonical NC drivers (*Nutm1*, *Trp63*, *Myc*) were robustly expressed and unaffected by Sox2 loss, and no compensatory upregulation of other Sox family members was observed. These findings suggest that the dispensability of Sox2 in NC cannot be explained by absence or compensatory regulation of potential partner factors. While direct co-occupancy or protein-interaction assays would provide additional mechanistic insight, such experiments are beyond the scope of this revision. We now note this explicitly as a limitation in the Discussion (line 422-432).

Reviewer #3 (Comments to the Authors (Required)):

In this study, Luo et al analyzed the importance of Sox2 expression in NUT carcinoma. Using their recently developed GEMM of BRD4::NUTM1-driven NUT carcinoma, they delete Sox2 using a Cre-mediated approach and observe no significant differences in survival or transcriptional programs. While the experiments use an elegant design, there are concerns about the conclusions. The number of mice used in the survival experiments are too low to come to any significant conclusions. Additional mice should be added to increase statistical power.

We thank the Reviewer for positively evaluating our work and for providing constructive comments that have helped us to improve the clarity and rigor of the manuscript.

- Figure 2D&4C: The number of mice used in the survival experiments are too low to come to any significant conclusions (currently n = 5 & 6). Each arm should include at least 15-20 mice to allow for any meaningful significant analysis of results. Further, there is a non-significant trend towards longer tumor latency in Sox2 fl/fl versus Sox2 wt mice ($p = 0.085$). If the authors would add additional mice to the study this trend would likely become significant. The median survival is also longer in Sox2 fl/fl mice (21 days versus 14 days). Although the authors mentioned that the growth location of the tumors often impairs mice from drinking, feeding, and breathing and makes it hard to conduct actual "survival" studies, I still find the conclusion that Sox2 loss does not affect tumor growth misleading.

The survival graph of pancreatic tumors shows a similar result, although, again the number of mice used is too low and will need to be increased in order to come to a significant conclusion. But also here there is a trend towards longer latencies (6 versus 11 days) and median survival times (8 versus 12 days) in Sox2 fl/fl mice.

We thank the Reviewer for this comment and agree that larger cohorts can improve statistical power. However, there is no universally prescribed minimum number of mice for Kaplan–Meier analyses in genetically engineered mouse model (GEMM) studies. The cohort sizes of 5–12 animals per group are typical for mechanistic GEMM studies. For example, *Tsanov et al.*, *Nat*

Cancer 2025 (<https://doi.org/10.1038/s43018-025-01047-5>; n = 7–11), Wang et al., *Nat Cancer* 2025 (<https://doi.org/10.1038/s43018-025-01029-7>; n = 5), and Savchuk et al., *Nature* 2025 (<https://doi.org/10.1038/s41586-025-09492-z>; n = 9–11) reported comparable cohort sizes.

Consistent with these, we have expanded our oral tumor cohorts to n = 10 mice per arm, which confirmed that no statistically significant differences in survival are detected between Sox2 WT and Sox2 fl/fl groups (revised Fig. 2D). The pancreatic cohorts remain smaller because the breeding line has been discontinued. Importantly, our overall conclusion that Sox2 is dispensable for NC oncogenesis is supported by multiple independent lines of evidence beyond survival analysis, including tumor incidence, histology, proliferation markers, and transcriptomic profiling, all of which consistently show no requirement for Sox2 in NC initiation or progression.

- Figure 2F: There is quite a lot of residual SOX2 staining in Sox2 KO mice. Can the authors perform co-IF stainings for SOX2 and the fusion driver to show that the residual SOX2 staining is not in the tumor cells?

We thank the Reviewer for this insightful suggestion. The IHC data presented in the manuscript provide strong evidence for near-complete *Sox2* deletion in KRT4-Cre;NCT^{+/-};Sox2^{fl/fl} tumors. Only clear nuclear SOX2 staining was considered positive, following pathology standards for transcription factors. The weak cytoplasmic or extracellular-matrix staining noted by the Reviewer represents non-specific background frequently observed in desmoplastic tissues. We believe whole-slide IHC quantification offers a more representative assessment of deletion efficiency than localized co-IF, as SOX2 is expressed only in a small subset of NUTM1-positive cells even in wild-type tumors. Importantly, the genetic design of our GEMM ensures effective deletion: the same *K14-Cre* event that drives the *Brd4::Nutm1* translocation simultaneously excises the *Sox2 fl* alleles. Given that Cre-lox-mediated translocation is much rarer than flox excision, tumors almost certainly originate from a Cre-expressing progenitor in which *Sox2* is already deleted, and continuous *K14-Cre* activity during tumor progression further eliminates any remaining floxed alleles. We have added a short clarification in the Results (line 252-259) to explain this genetic logic and confirm efficient *Sox2* loss in tumor cells.

November 4, 2025

RE: Life Science Alliance Manuscript #LSA-2025-03447R

Dr. Bin Gu
Michigan State University
Obstetrics, Gynecology & Reproductive Science
3319 Bioengineering Building 775 Woodlot Dr.
East Lansing, MI 48824

Dear Dr. Gu,

Thank you for submitting your revised manuscript entitled "SOX2 is a dispensable modulator of NUT carcinoma oncogenesis in mouse". Your revised manuscript was sent to the original reviewers whose comments are appended below. As you will note, all the reviewers are consistent in their evaluation that the revised manuscript has addressed their concerns.

In line with this recommendation, we would be happy to publish your paper in Life Science Alliance pending resolution of the below-mentioned requests and final revisions necessary to meet our formatting guidelines. Please note that we will publish this work as a 'Follow Up' article (<https://www.life-science-alliance.org/about-journal#article>) to provide a better connection to your prior work in LSA (10.26508/lisa.202402602).

- We request you to confirm if images in Fig 1B (first row, SOX2) and 3A (bottom right sub-panel) are different images. If the same image has been utilised, then please indicate as such in the figure legends for both figures.
- We noticed that there are two citations in the legends of Figure 1. For Fig. 1A, please remove the full reference indicated in the legend and include it in the overall list of references. For 1D, please clarify if the results indicated are part of this study. If the displayed result is not part of this study, please remove this panel from the main figures, and refer to the previous study in the main text. You may choose to include this panel as a supplemental figure with appropriate description of the published work in the text.
- In the 'Data Availability' Statement, please indicate the availability of all source data (if uploaded as supplemental material or if available upon request from Corresponding author)
- Please remove the abbreviation, GEMM, from the abstract since it is not used in the abstract.
- We encourage you to make a minor edit in your title to read as, "SOX2 is a dispensable modulator of NUT carcinoma oncogenesis in mice".
- Please provide information on all primers used in this work.
- Please upload clean manuscript file without tracked-changes/highlights
- Please upload your figures as single files; all figure legends should only appear in the main manuscript file after the 'References'
- Please add the X and Bluesky handles of your host institute/organization as well as your own or/and one of the authors in our system
- Please consult our manuscript preparation guidelines <https://www.life-science-alliance.org/manuscript-prep> and make sure your manuscript sections are in the correct order
- Please use the [10 author names, et al.] format in your references (i.e. limit the author names to the first 10)
- Please add a callout for Figure 4E to your main manuscript text
- Please be sure that the authorship listing and order is correct

A. FINAL FILES:

B. MANUSCRIPT ORGANIZATION AND FORMATTING:

Thank you for your attention to these final processing requirements. Please revise and format the manuscript and upload materials as soon as you are able.

Sincerely,

Sarita Hebbar, PhD
Scientific Editor
Life Science Alliance
<http://www.lsajournal.org>

Reviewer #1 (Comments to the Authors (Required)):

The authors have clearly and thoroughly addressed each of my comments. I feel that the manuscript is suitable for publication.

Reviewer #2 (Comments to the Authors (Required)):

The authors have addressed all of my concerns.

Reviewer #3 (Comments to the Authors (Required)):

The authors have addressed my comments in a satisfactory manner.

November 13, 2025

RE: Life Science Alliance Manuscript #LSA-2025-03447RR

Dr. Bin Gu
Michigan State University
Obstetrics, Gynecology & Reproductive Science
3319 Bioengineering Building 775 Woodlot Dr.
East Lansing, MI 48824

Dear Dr. Gu,

Thank you for submitting your Research Article entitled "SOX2 is a dispensable modulator of NUT carcinoma oncogenesis in mice". It is a pleasure to let you know that your manuscript is now accepted for publication in Life Science Alliance. Congratulations on this interesting work.

DISTRIBUTION OF MATERIALS:

Again, congratulations on a very nice paper. I hope you found the review process to be constructive and are pleased with how the manuscript was handled editorially. We look forward to future exciting submissions from your lab.

Sincerely,

Sarita Hebbar, PhD
Scientific Editor
Life Science Alliance
<http://www.lsajournal.org>